# Spatio-temporal regularization for deep MR Fingerprinting

**Mohammad Golbabaee**[1]                                        M.GOLBABAEE@BATH.AC.UK

**Dongdong Chen**[2]                                                    D.CHEN@ED.AC.UK

**Mike Davies**[2]                                                    MIKE.DAVIES@ED.AC.UK

**Marion I. Menzel**[3,4]                                                  MENZEL@GE.COM

**Pedro A. Gómez**[3,4]                                          PEDRO.GOMEZ@TUM.DE

[1] *Computer Science department, University of Bath* [2] *School of Engineering, University of Edinburgh*
[3] *School of Bioengineering, Technische Universität München,* [4] *GE Healthcare*

## Abstract

We study a deep learning approach to address the heavy storage and computation requirements of the baseline dictionary-matching (DM) for Magnetic Resonance Fingerprinting (MRF) reconstruction. The MRF-Net provides a piece-wise affine approximation to the (temporal) Bloch response manifold projection. Fed with non-iterated back-projected images, the network alone is unable to fully resolve spatially-correlated artefacts which appear in highly undersampling regimes. We propose an accelerated iterative reconstruction to minimize these artefacts before feeding into the network. This is done through a convex regularization that jointly promotes spatio-temporal regularities of the MRF time-series.

## 1. Deep Bloch manifold projection

Magnetic Resonance Fingerprinting (MRF) (Ma et al., 2013) recently emerged to accelerate the acquisition of tissues' *quantitative* NMR characteristics. Dictionary-matching (DM) approaches proposed for the MRF reconstruction do not scale well to the complexity of the emerging multi-parametric quantitative MRI problems. Deep learning (DL) methodologies have been recently introduced to overcome this problem (Cohen et al., 2018; Virtue et al., 2017; Golbabaee et al., 2019a). Time-series of Back-Projected Images (BPI) are fed into a compact neural network which temporally processes voxel sequences and approximates the DM step to output the parametric maps. For instance, our proposed MRF-Net (Figure 1(a)) is able to accurately approximate the DM step by saving more than 60 times in memory and computations (Golbabaee et al., 2019a). The MRF dictionary is only used for training and not during parameter recovery. Figure 1(b) shows that the network provides a piece-wise affine approximation to the Bloch response manifold projection and that rather than memorizing the dictionary, it efficiently clusters this manifold and learns a set of hierarchical *matched-filters* for affine regression of the NMR characteristics in each segment (for more details see (Golbabaee et al., 2019a)).

## 2. Our parameter estimation pipeline

Trained by independently corrupted noisy fingerprints, the MRF-Net acts only along the temporal domain and is unable to correct for dominant spatially-correlated (aliasing) arte-

facts appearing in highly undersampled regimes. Also larger convolutional models aiming to learn spatio-temporal structures are prone to overfitting due to the limited access to properly large ground-truth parametric maps in practice. Further, such approaches build customized de-noisers which require expensive re-training by changing sampling parameters i.e. the forward model. We address these shortcomings by taking a dictionary-free compressed sensing approach to spatio-temporally process data before feeding into the compact and easily-trained MRF-Net. The undersampled k-space measurements $Y \in \mathbb{C}^{m \times L}$ acquired across $L$ timeframes are first processed by solving the following convex regularized problem (Golbabaee et al., 2019b):

$$\widehat{X} = \arg\min_X \|Y - \mathcal{A}(XV_s^H)\|_2^2 + \lambda \sum_{i=1}^{S} \|X_i\|_{TV} \quad \text{(P1)}$$

in order to find $S \ll L$ subspace images $X \in \mathbb{C}^{n \times S}$. A MRF dictionary $D \in \mathbb{C}^{L \times d}$ ($L \ll d$) is used during the training phase for (unsupervisedly) learning the $S$ principal subspace bases $V_s \in \mathbb{C}^{L \times S}$ for dimensionality reduction. The forward operator $\mathcal{A}$ models the multi-coil sensitivities and the per-frame subsampled 2D Fourier Transforms. The low-rank subspace model is a convex (in fact linear) relaxed representation of the temporal dictionary responses and when accurate enough, it is computationally advantageous over the full image representation $X^{Full} \approx X^s V_s^H \in \mathbb{C}^{n \times L}$ because it reconstructs smaller objects and promotes temporally low-rank structures (Assländer et al., 2018). This prior alone is, however, insufficient to obtain artefact-free solutions e.g. when using spiral readouts (Golbabaee et al., 2018). We additionally use the Total Variation (TV) regularization to promote spatial smoothness across recovered subspace images. (P1) can be efficiently solved using FISTA algorithm (Beck and Teboulle, 2009).

## 3. Numerical Results

Methods are tested on a simulated brain phantom[1] and a healthy human brain acquired using the Steady State Precession (FISP) sequence in (Jiang Y et al., 2015) and spiral readouts which sample $m = 732$ k-space locations in each of the $L = 1000$ time-frames in order to reconstruct $n = 256 \times 256$ resolution parametric $T1$ and $T2$ maps. We simulate $d$=113'640 fingerprints and use the clean temporal responses for unsupervised subspace model learning of sufficiently low-rank ($S$=10). Further, fingerprints corrupted by additive white Gaussian noise (data augmentation by factor 100) supervisedly train the dimension-reduced MRF-Net on a standard CPU desktop.

We compare three methods for reconstructing subspace images before feeding to the MRF-Net: non-iterative BPIs i.e. $\widehat{X} := A^H(Y)V_s$, and iterative reconstructions incorporating ii) only the low-rank (LR) subspace prior by solving (P1) with $\lambda = 0$, and iii) joint TV and subspace spatio-temporal priors (LRTV) by solving P1 with an experimentally tuned $\lambda = 2 \times 10^{-5}$. Note that the BPIs are the first iteration of the LR. Figure 2 shows the reconstructed maps. Undersampling artefacts are visible in BPI+MRF-Net. The subspace iterations of LR+MRF-Net also admit undesirable solutions with high-frequency artefacts due to the insufficient measurements collected from the k-space corners in spiral readouts

---

1. http://brainweb.bic.mni.mcgill.ca/brainweb

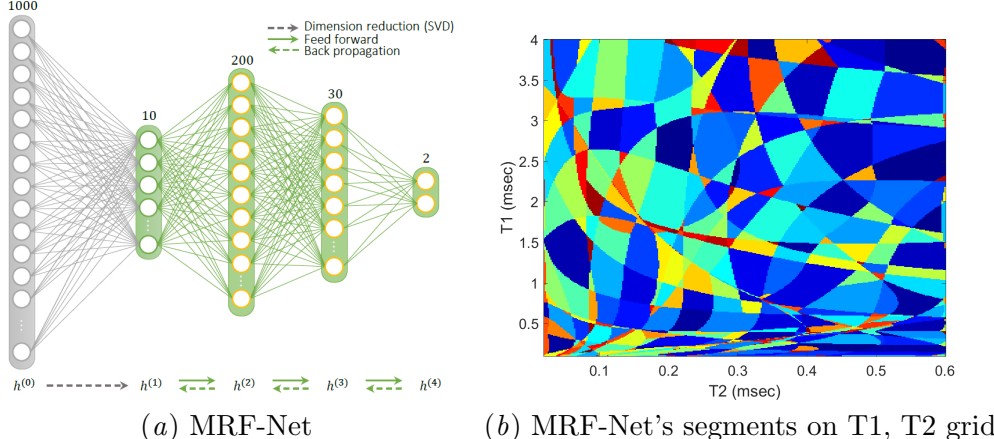

(a) MRF-Net

(b) MRF-Net's segments on T1, T2 grid

Figure 1: (a) Illustration of the MRF-Net (Golbabaee et al., 2019a): Inputs $h^{(1)}$ are the voxel sequences of the subspace image reconstructed by (P1) and outputs $h^{(4)}$ are the per-voxel T1 and T2 parameters. The MRF-Net has implicitly 4 layers by including the unsupervisedly learned subspace projection (first layer in gay) incorporated in solving (P1). Three last layers use nonlinear ReLU activations (orange) and are supervisedly trained by standard backpropagation to approximate subspace dictionary matching. (b) MRF-Net hierarchically segments the input space and learns a piece-wise affine mapping between input-outputs for each segment. The end-to-end segments are shown on the T1, T2 grid that generated the Bloch response manifold.

(for details see (Golbabaee et al., 2018)). By adding sufficient spatial regularization, the proposed LRTV+MRF-Net outputs artefact-free maps within 8-12 iterations.

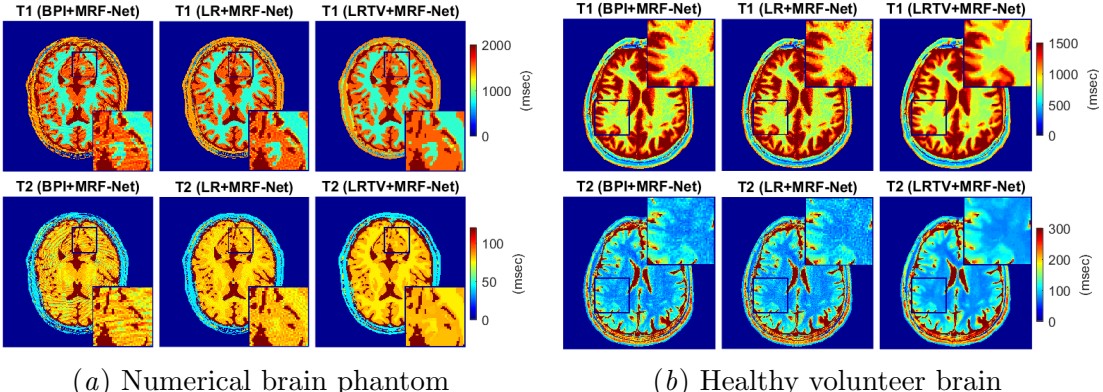

(a) Numerical brain phantom

(b) Healthy volunteer brain

Figure 2: Reconstructed T1 (top row) and T2 (bottom row) maps for numerical brain phantom and healthy volunteer data using MRF-Net fed with the non-iterated BPIs (left column), iteratively reconstructed images with only low-rank (LR) subspace prior (middle column), and iteratively reconstructed images with joint TV and low-rank subspace (LRTV) priors (right column). Undersampling artefacts are visible in BPI+MRF-Net. The subspace iterations of LR+MRF-Net removes them but it admits a solution with high-frequency artefacts due to the insufficient measurements from the corners of the k-space. By adding sufficient spatial regularization, the proposed LRTV+MRF-Net outputs artefact-free maps.

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
