# OpenReview forum: "Spatio-temporal regularization for deep MR Fingerprinting"
_MIDL.io/2019/Conference/Abstract — MIDL Abstract 2019_

### Official Review · AnonReviewer1 · 2019-04-29
**Substantial overlap with previous conference presentation**

**Rating:** 2
**Confidence:** 2

**Review:**

Authors are combining convex regularization with a deep neural network to achieve efficient T1/T2 parameter mapping based on MR Fingerprinting. I thought the work itself was quite interesting and within the scope of MIDL. Critical comments are that I wondered how the authors can claim "artifact-free reconstruction" without comparing their results to a ground truth (which, at least in case of the simulated data, should be readily available). It was also not clear to me what I was supposed to learn from Fig. 1 (b). By themselves, these points would not lead me to recommend rejection as an extended abstract.

However, reviewers of the extended abstracts were instructed to accept submissions that are based on recent journal papers, but to reject ones that were previously presented at other conferences in substantially similar form. It is my impression that the latter is the case here. It did not become clear to me what distinguishes this work from the 2019 ISMRM abstract by the same authors on the same topic.

---

### Official Review · AnonReviewer2 · 2019-04-30
**Two step regularization based on Total Variation and deep learning for MR fingerprinting; claims of artefact free reconstruction**

**Rating:** 3
**Confidence:** 1

**Review:**


Summary:

A deep learning method to perform spatio-temporal regularization is presented in this work. As magnetic resonance fingerprinting (MRF) relies on performing computationally expensive dictionary matching, a deep learning method (MRF-Net) is proposed to approximate the temporal responses. Further, an iterative reconstruction step based on low rank total variation (LRTV) subspace priors are used to enforce spatial regularization.

Comments:

+ Experiments on simulated brain phantom and human brain show that MRF-Net input with LRTV based reconstructions does reduce artefacts to a large extent
+ Comparison with other reconstructions (non-iterative and only LR) highlight the influence of spatial regualrization

- Visual inspection does reveal a smoothened output for LRTV+MRF-Net. However, can the claim that these reconstructions are artefact-free be quantified?

---

### Decision · Program_Chairs · 2019-05-06
**Acceptance Decision**

Accept